# Analyzing Escalator Infrastructures: A Pilot Study in Santiago Metro

**Ariel López [1], Anibal Tapia [2] and Sebastian Seriani [3,***

[1] Facultad de Arquitectura y Urbanismo, Universidad de Chile, Santiago 8330015, Chile; ariellopez@ug.uchile.cl
[2] Facultad de Ingeniería y Ciencias Aplicadas, Universidad de los Andes, Santiago 7550000, Chile; aetapia@miuandes.cl
[3] Escuela de Ingeniería de Construcción y Transporte, Pontificia Universidad Católica de Valparaíso, Valparaíso 2362804, Chile
* Correspondence: sebastian.seriani@pucv.cl

**Abstract:** The behavior of passengers in urban railway stations (i.e., metro stations) is dependent on environmental, cultural, and temporal factors. This affects how escalator infrastructures are used, with some passengers standing on one side of the steps to allow others to walk and, in other cases, others standing or walking on both sides of the steps. In this research, escalator infrastructures were studied to better understand the relationship between different conditions and passenger behaviors through a method based on video cameras, passenger detection techniques, and a simulation framework. The results indicate that passenger behavior could affect escalator infrastructure as it varies significantly depending on the time of day, type of station, local culture, and other environmental factors. The implications of these findings for the planning and design of the infrastructure of railway stations are discussed, and practical recommendations are proposed to improve the efficiency of escalator usage in urban railway stations.

**Keywords:** passenger; behavior; escalator; infrastructure; urban railway station

## 1. Introduction

The Santiago Metro is an underground transportation system that connects 32 boroughs in Santiago, Chile, with a total of 7 operational lines and 136 stations distributed across its different lines and a total length of approximately 140 km, which in 2019 transported a maximum of 695 million passengers [1]. The Santiago Metro has been recognized as one of the most modern and efficient underground transportation systems in Latin America, standing out for its punctuality, safety, and comfort for passengers [2]. The continuous expansion of the network, with the implementation of new lines, has allowed for an expansion of services, offering access to a greater number of users. It is expected that even more lines will be inaugurated in the near future, indicating that demand for the Santiago Metro will continue to increase.

Each day, more than 2.5 million trips are registered in the urban railway network of Santiago, generating high interaction and congestion levels, which affect the infrastructure performance of escalators inside the stations [3]. A study carried out by the Moovitapp App revealed that, on average, a passenger spends 84 min a day on the Santiago Metro and that 74% of passengers transfer to other modes of transport at least once a day [4].

The high interaction and congestion levels are generated especially at connection points between different spaces in urban railway stations: the train platform, the platform–stair area, mezzanines, complementary spaces (for example, commerce), and the city (street level). However, all these spaces have connections through stairs or escalators, which may cause bottlenecks and evacuation problems, affecting the capacity of the whole station [5].

Infrastructures such as escalators are intended to speed up the evacuation process of passengers, connecting different spaces within the station and to the city. However, these

infrastructures may lose their efficiency when a high agglomeration of passengers is reached at the base of the escalator or when there is inappropriate passenger behavior, affecting the use of the infrastructure and generating crossings of pedestrian flows or accidents. In terms of safety, the risks of accidents due to crowding are becoming increasingly higher considering the constant increase in population that the city is experiencing. Therefore, the accident rate on escalators has been increasing, which has been a topic of interest for the Metro company to ensure the well-being of users [6].

To solve this, different measures have been applied in urban railway stations. In the case of the London Underground, signaling with audible and visual information is used to indicate that passengers must keep to the right when moving up through the escalator, generating the possibility for those who want to move faster to overpass those passengers from the left-hand side (see Figure 1a) [7]. In the case of the Santiago Metro, there have been some attempts to solve this type of problem. Some stations such as Plaza Egaña (Line 4) have signs located on the ceiling of the station and markings on the stairs of the escalator to indicate where passengers should be located when evacuating the station (see Figure 1b).

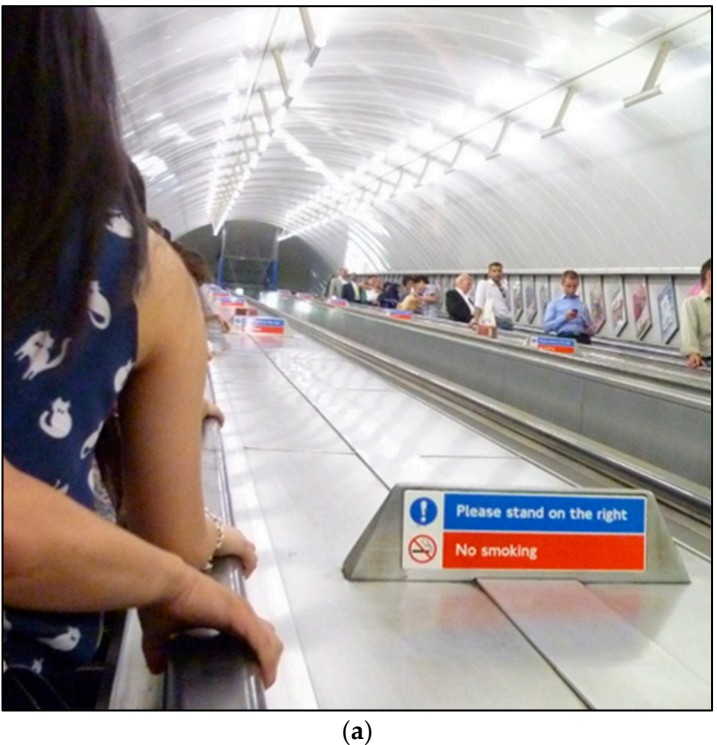
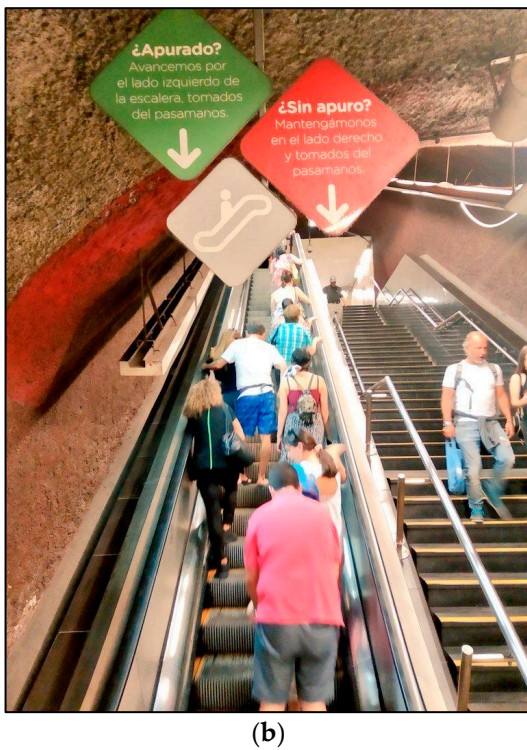

| (a) | (b) |
|---|---|

**Figure 1.** Escalators in urban railway stations: (**a**) the London Underground (sign used: "please stand on the right") and (**b**) the Santiago Metro (sign used: "in a hurry?" or *¿apurado?* and "not in a hurry or *¿sin apuro?*".

It is well known that the behavior of passengers in urban railway stations is affected by different factors such as the physical design, people around, information, and environment [8]. Taking this sentence into the context of the Santiago Metro, the situation is as follows: passengers are not used to standing on the right-hand side of the escalator, which affects those passengers moving at a higher speed or needing more space to evacuate the station. Therefore, the effect of these types of solutions on the capacity of the escalator is unknown, and it is not possible to understand their relationship with passenger behavior.

For this study, the general objective was to study the effect of passenger behavior on the escalator infrastructure in urban railway stations, taking the Santiago Metro as a case study. To this, the following specific objectives are defined: (a) define the relevant variables that affect the behavior of the escalator infrastructure in urban railway stations; (b) observe the behavior of passengers on escalators in existing stations through video cam-

eras and passenger detection techniques; (c) understand the relationship between different conditions and passenger behaviors using a simulation framework; and (d) propose design recommendations to improve the efficiency of escalator usage in urban railway stations.

The article is organized in the following manner. Section 2 describes the studies related to escalators to account for the existing literature. Then, in Section 3, the method used based on observation is defined to later present the results obtained in Section 4. Finally, in Section 5, the conclusions are described.

## 2. Existing Studies on Escalators in Urban Railway Stations

One way to reduce travel times and improve the user experience in urban railway stations is by reducing the level of crowding generated at the beginning of the escalator. At the beginning of the escalator, a bottleneck is generated, and therefore the capacity of the infrastructure is reduced, which is related to different factors such as the width of the escalator, the location of the escalator (e.g., closer to a wall or next to a corridor inside the station), and the behavior of the passengers [9]. These authors state that the behavior in this area of the escalator is an important factor for estimating the capacity of the escalator. The authors proposed a model to influence the movement of passengers, increasing the efficiency of the passenger flow and reducing travel times through passageways, normal stairs, and escalators. In the same way, the authors reported that the design of public transport facilities is the most important variable to consider in the behavior of passengers and that a correct prediction is key in the design due to its impact on the entry and exit flows on escalators.

A study carried out by Davis and Dutta [10] on the London Underground included passenger behavior in order to estimate escalator capacity in passengers per minute. With the implementation of the model reported by Mayo [11], the authors studied the behavior of passengers standing and walking on an escalator. Parameters such as the speed of the escalator, its length, and the flow of passengers per hour were included. In addition, the behavior of passengers was studied considering the rise of the escalators, the corner effect in front of the facility, and the presence of multiple escalators. The authors reported that the capacity of the escalator depends on the speed of the escalator measured in feet per minute, the height of the escalator measured in feet, and the flow of passengers per hour. As part of the analysis, different ways of calculating the capacity based on two types of behavior that passengers can show on an escalator are disclosed by [10]. The first type of behavior is described when passengers are standing on the escalator. The model described is based on the speed of the escalator (meters/minute), the width of the steps of the escalator (meters), the number of steps on the escalator per minute, the proportion of steps used while standing on the escalator, and the number of passengers on the escalator per minute. On the other hand, the second form of behavior is related to the passengers who travel on the left side of the escalator and those who stand on the right side of the steps. This behavior is described by the relationship between the effective speed of the passengers walking on the escalator (meters/min), the speed of the passengers walking on the escalator (meters/minute), the number of effective steps on the escalator per minute, the proportion of steps used when walking on the escalator, and the number of passengers on the escalator per minute.

The authors [10] concluded that the longer the escalator, the fewer people are willing to walk on it and, therefore, a greater number of standing passengers are registered. In addition, the authors found that the capacity of the escalator is related to the way that passengers get to board the escalator, that is, from the front (e.g., when the escalator is located next to a corridor inside the station) or from the sides of the escalator (e.g., when the escalator is located closer to a corner inside the station).

Although the study by Davis and Dutta [10] defined the factors that most impact passenger behavior, it was carried out on the London Underground where there are culture and behavior patterns different from those that can be generated in stations in Santiago. This behavior alluded to the fact that people lined up on one side of the escalator entrance

(producing a bottleneck) to be able to leave standing on the right in order not to interrupt the passage of people who wanted to walk on the left of the escalator. According to Al-Sharif [12], behavior on escalators should be taught in schools through educational programs such as the Safe-T-Riders program. Therefore, this information is taken as a precedent, but it is subject to comparison with the circumstances that need to be visualized in field observations considering the local conditions of each culture to see similarities and differences.

In relation to safety, there are certain factors that affect or cause accidents on escalators such as the escalator's design, flow management, and passenger behavior [12]. Although escalator design and flow management are two important factors, the behavior of passengers on the escalator is the biggest cause of accidents. Within this factor, the three most common risk situations are identified: reading while traveling, traveling without looking ahead, and standing on the wrong side. It should be noted that standing on the right side of the escalator is not only a cultural factor but also acts as a safety method for passengers to avoid falling on the escalator or from the escalator. This is explained as follows: if a person stands to the left of the escalator and remains static, the person behind may want to move, creating a risky situation, which includes a possible accident. In this way, standing on the correct side of the escalator (to the right) works as an accident minimizer. In summary, the author [12] proposes an accident management model that includes five stages: alarm activation, operation decision, inspection and analysis, adjustment to maintenance and inspection, and finally the modification or adjustment to the escalator's design.

With respect to other variables, Fruin [13] reported the level of service (LOS) of different infrastructures such as stairs and escalators in urban railway stations. The author reported that the speed of passengers in both directions was measured, with them finding that the average ascent speed is in a range of 0.13 to 0.75 m/s for the first levels and then it increases to a range of 0.36 to 0.76 m/s. The author also reported the fundamental relationships between density and speed, where it is established that, while the density is lower, the higher the maximum speed that the passenger can reach is (and, on the contrary, while the density is higher, the lower the speed reached by the passenger is). Lazi, et al. [14] used the LOS to assess the quality of service presented on the escalators of different places in Singapore, such as urban railway stations. A relevant point for Lazi et al. [14] is that an escalator provides a series of benefits to passengers that go beyond achieving a vertical connection between one place and another, but rather, it is an element that helps the quality of life of people by reducing travel time and energy consumption among other benefits.

Taking into account the benefits of escalators, the Santiago Metro is promoting the use of these elements, which makes it possible to connect spaces on different levels with a slope of at most 70% and a speed of 0.5 m/s, which could move approximately 60 passengers/minute/meter [15]. However, according to Kahal and Rastogi [16], the capacity of an escalator is influenced by other factors such as culture, whereby passengers may keep a high distance between them or walk at a slower speed, occupying more steps on the escalator. If the escalator is reaching capacity, passengers may choose other options such as normal stairs or elevators, whereby passengers are not only deciding to use the escalator based on travel time but also on comfort and safety [17]. Therefore, it is important to understand the decisions/choices of passengers before using simulation tools to study the capacity of escalators [18]. For example, Kinsey et al. [19] state that human factors and escalator strategies have an influence on decision-making and, therefore, the evacuation of escalators. The authors used EXODUS evacuation software based on the London Underground system. More recent studies [20] have used the cellular automata approach to simulate passengers on an escalator, considering passengers walking, standing, or forming groups.

Research related to passenger behavior in subway stations, specifically regarding the choice between stairs and escalators, has seen significant advancement in recent years. A recent study by Zhang et al. [21] shed light on passenger selection behavior in this context. Furthermore, Li et al. [22] explored the impact of height on pedestrian route selection in these stations, while Wei et al. [23] focused on the optimized design of security systems. Additionally, Li et al. [24] modeled passenger group panic in emergency situations, Zhao

et al. [25] investigated passengers' dependency on station functions, and Su et al. [26] developed calculation methods for passenger flow distribution. On the other hand, Lee et al. [27] analyzed subway passengers' route preferences, while Yang et al. [28] concentrated on passenger evacuation path planning in emergency situations. Furthermore, Guo and Zhang [29] worked on the real-time monitoring of passenger flow congestion, and Mo et al. [30] addressed the calibration of simulation models using smart card and train movement data.

Moreover, in relation to agent-based pedestrian models, Wang et al. [31] analyzed pedestrian behaviors in metro stations. Their model is able to reproduce various pedestrian behaviors and has been applied to evaluate the design of metro stations. The study demonstrates the flexibility and effectiveness of using simulation models to understand and optimize pedestrian flow in metro stations. Another study by Wang et al. [32], discusses the analysis and optimization of passenger flowlines at a high-speed railway station using Anylogic simulation software version 8.1. The simulation model of passenger flowlines was established to simulate passenger flow during peak and off-peak hours. The study demonstrates the application of simulation models in optimizing passenger flow and improving the operation capacity of railway stations. In addition, Li et al. [22] conducted a study on passenger flow simulation in urban subway stations using Anylogic. The simulation model based on the theories of agent and queuing system was used to acquire the results and evaluate the operational efficiency of subway stations. This study highlights the use of simulation models to attract more passengers and improve the service quality of subway stations.

Collectively, these research endeavors have provided a comprehensive understanding of how factors such as height, security systems, panic, and route preferences can influence passenger behavior in subway stations. This body of work offers a solid foundation for informed decision-making by subway system operators and designers in terms of station design, crowd management, and emergency preparedness, with the aim of enhancing efficiency, safety, and the overall passenger experience within these systems.

Despite the important research carried out, new studies are needed to better represent the behavior of passengers and their effect on the capacity of escalators, which is the main objective of this paper.

## 3. Method

The method used in this study is composed of three steps. Figure 2 shows a diagram of this method which is based on observation in existing stations.

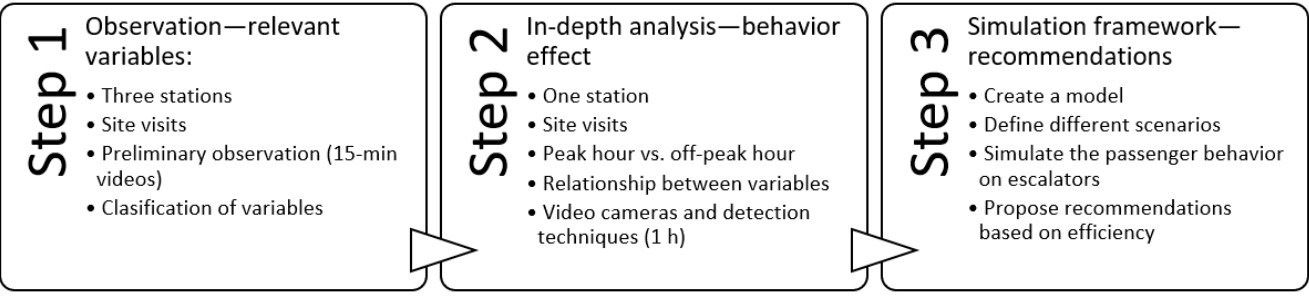

**Figure 2.** Method based on observation in existing stations to study the effect of passenger behavior on escalator infrastructure in urban railway stations.

### 3.1. Step 1: Observation—Relevant Variables

Site visits and preliminary observations were carried out on the Santiago Metro during different periods of time (e.g., 15 min during the peak hour or off-peak hour) at urban railway stations to select the variables that affect the behavior of passengers on escalators.

To this, a sample of stations where escalators are used was chosen on Line 1 of the Santiago Metro (which is the oldest and busiest line in the network): Los Domínicos, Manquehue, and Tobalaba.

To analyze the behavior in those stations, video cameras were used at a point where the number of passengers entering the escalator per unit of time could be observed (passengers/min). For example, Figure 3 shows the different views of the cameras used in Tobalaba station (Figure 3a) and Manquehue station (Figure 3b).

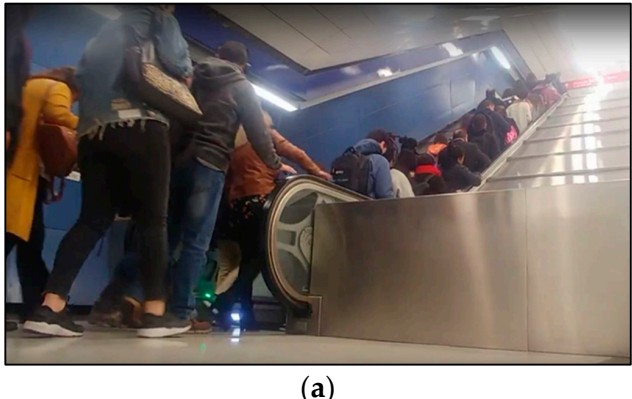
(**a**)

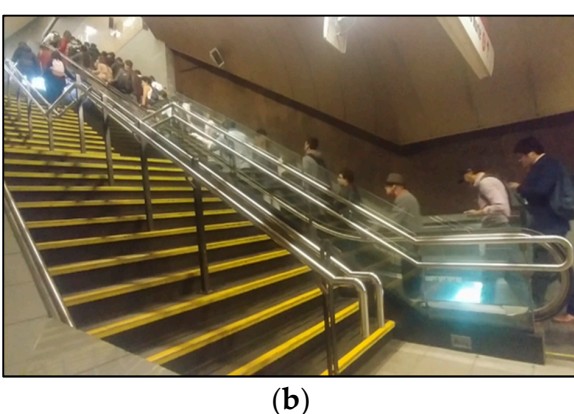
(**b**)

**Figure 3.** Video camera position used on the escalators: (**a**) Tobalaba station and (**b**) Manquehue station.

### 3.2. Step 2: In-Depth Analisis—Behavior Effect

Site visits and in-depth analyses were conducted to observe the relevant variables (defined in Section 3.1) for a longer period in another station. To this, Universidad de Chile station was selected, which is located in a central part of the city with an influx of passengers with different purposes depending on the time of day since this station provides access to office spaces, services, shops, universities, and schools. In this case, the escalators that connect the mezzanine level with the Ahumada pedestrian street were observed. This access has four escalators that generally operate as two entering the station and two exiting the station. Unlike the previous observations in Section 3.1, on this escalator, artificial vision software was used to complement the observation and the analysis of the information. To this, video cameras were used to record the movement of passengers at different times of the day, using detection techniques (Opensource Software Datafromsky, https://datafromsky.com/,access date: 2 August 2023). The detection of passengers and their trajectories were recorded during three periods of time:

- Morning (7:30 a.m.–8:30 a.m.), which is the time that people leave through this access to go to work and students go to school.
- Midday from 1:00 p.m. to 1:30 p.m., which is the lunch hour.
- Late rush hour (5:30 p.m.–6:30 p.m.), which is the departure of people from their jobs and of schoolchildren from school.

Therefore, these periods of time are related to the trip's purpose and the type of passengers who use the escalator.

### 3.3. Step 3: Simulation Framework—Recommendations

Based on Universidad de Chile station, a simulation model was developed using the open-source software Anylogic to simulate different escalator configuration scenarios to improve the efficiency of usage of the escalator [33]. The simulation scenarios considered the relevant variables that affect the behavior of passengers.

Anylogic unifies all modern modeling and simulation approaches by simultaneously integrating agent-based simulation, system dynamics, and discrete event systems; with macro, meso, and microscopic scale levels. It allows one to work with flowcharts, state diagrams, action diagrams, and process flow diagrams. It operates in different environments and can be integrated into Java engines and editing tools. It allows the export of simulation

models to small Java applications to be used from any web browser. Additionally, it has a pedestrian library that describes pedestrian behavior, and a railway library that allows for programming the operation of trains. Both pedestrians and trains can be visualized in 3D. Therefore, Anylogic has all the necessary characteristics to carry out the simulations of this research.

The Anylogic pedestrian library used allows one to simulate the pedestrian flows in physical environments such as an escalator in urban railway stations. It allows the collection of statistical data such as the pedestrian density. It is a powerful library to detect potential problems that arise from the relationship between pedestrians and the environment and other pedestrians [33].

The model has two design areas: environment and behavior. The environment incorporates the escalator and its surroundings (e.g., walls, areas, and stairs). Behavior defines the physical rules of pedestrian behavior.

In this framework, Anylogic allows pedestrians to be programmed with a high degree of artificial intelligence. This study is limited to behaviors and decision-making inside an urban railway station; it will be limited to programming behavioral variables such as maximum speed, free flow speed, tolerance levels to obstacles and other pedestrians, influx rates, and behavior on the escalator, among others [33].

The behavior of passengers in this framework is defined as two physical rules: walk on the left side of the escalator and stand on one or both sides of the escalator [33].

As a result of the simulation, different recommendations are discussed. These recommendations are based on issues such as crowd management strategies (e.g., queue management at the entrance of the escalator) and operation of the escalator (e.g., the speed of the escalator), which could have an impact on the efficiency of use.

It is important to mention that the model was run in Anylogic Cloud, version Anylogic Personal Learning Edition @{buildId} (Java version: 11.0.16), using the platform of Linux 5.15.0-1028-aws 64-bit.

## 4. Results

The results of the observations and simulations are presented in the following sections. These results are reported according to the method defined in the previous section.

### 4.1. Exploring the Relevant Variables That Affect the Behavior of Passengers

From the observations explored in the three stations on Line 1 (Los Dominicos, Manquehue, and Tobalaba), it can be reported that the relevant variables that affect the capacity of escalators are:

- The density of passengers on the escalator which can be affected by the level of demand at the station [passengers/m$^2$].
- The flow of passengers on the escalator [passengers/min].
- The behavior of passengers, which depends on different variables:
  - The length of the escalator, defined as the physical length of the escalator [m].
  - The location of the passengers on the escalator (e.g., if the passengers are positioned to the right or left of each step of the escalator).
  - The manner in which passengers arrive at the escalator, i.e., whether they arrive from the front or from the side (e.g., from a transfer corridor or next to a corner from the mezzanine space)
  - Whether there is congestion or a bottleneck (e.g., passengers forming lines of flow to board the escalator).

It is important to mention that the capacity of the escalator is also affected by the speed of the escalator. However, the speed of the escalator was not considered in this analysis, since its value is within a range of 0.50 and 0.65 m/s [34], and therefore, it is considered a fixed variable for all escalators of the Santiago Metro which depends on the conditions of the designer or manufacturer.

These variables are in concordance with the literature review presented in [10,14,18]. As a summary, Table 1 presents the variables defined previously, which are explained for each of the three stations observed on the Santiago Metro.

**Table 1.** Summary of observed behavior on escalators in the Los Dominicos, Manquehue, and Tobalaba stations.

| Station | Passenger Flow [ped/min] | Length of Escalator (m) | % Standing Passengers on the Right Side | % Walking Passengers |
|---|---|---|---|---|
| Los Dominicos | 22 | 13 | 83% | 17% |
| Manquehue | 57 | 10 | 83% | 17% |
| Tobalaba | 50 | 6 | 83.5% | 16.5% |

- Los Dominicos station: This is the terminal station of Line 1, which is located in the eastern sector of the capital and connects users of the metro network with their workplaces, places of study, among others. The observed passenger flow was 22 passengers/min, which is considered low since an escalator can reach a capacity of 60 passengers/min. This observation was made during 15 min in the off-peak hour (12:00 p.m. to 12:15 p.m.) and therefore during the period of low passenger demand. Regarding the factors that affect capacity, the length of the stairs was measured, with them reaching 13 m. Likewise, the passengers were distinguished according to their location on the stairs, that is, those who remained standing (83%) and those who walked along the escalator (17%). In relation to the way passengers reach the escalator, it was observed that lines of flow are not generated when boarding, which produces correct transit without generating traffic jams. Finally, no passengers with reduced mobility were observed, since they prefer to use the elevators located in the station. Passengers must board the escalator after leaving the exit doors from the platform, for which they must make a movement in the shape of a "U" since the escalator is located near a corner of the station (see the diagram in Figure 4).

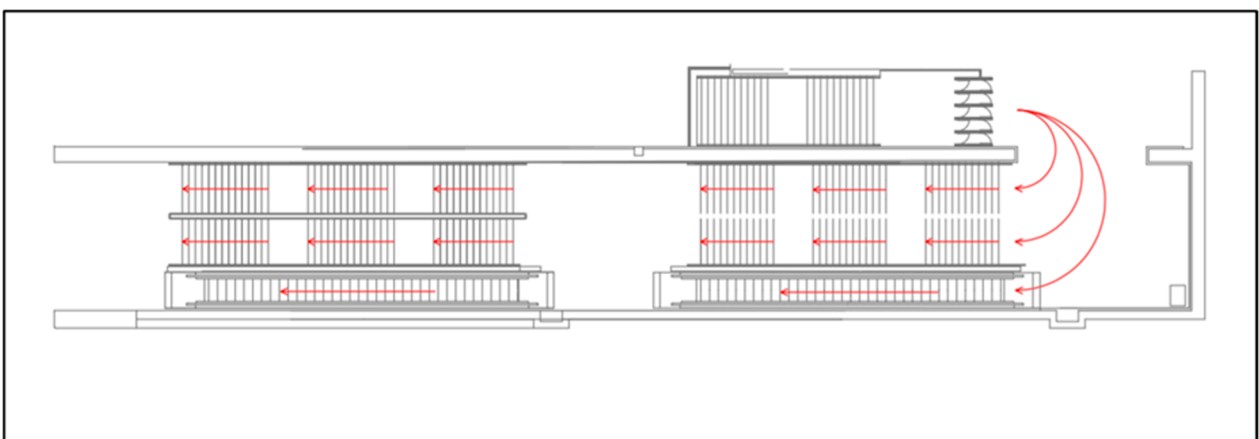

**Figure 4.** Passengers boarding in a "U" turn manner at the escalator in Los Dominicos station (red arrows represent the direction of the flow to exit the station through the escalator).

- Manquehue station: In the case of Manquehue station, the exit to Rosario Norte Street was observed, where the flow on the escalator was recorded (see Figure 3b in Section 3.1). It is important to note that Manquehue station is characterized by being located in a central location in the Las Condes commune where it allows users to connect with different places of interest such as their workplaces, homes, schools, universities, and shopping centers, among others. Observing the flow of passengers (passengers/min) that board the escalator, it was possible to obtain a flow of

57 passengers per minute, which was measured during 15 min in the morning rush hour (8:00 a.m. to 8:15 a.m.). Regarding the factors that affect capacity, it was possible to establish that the length of the escalator is 10 m. In the same way, as in Los Dominicos station, the location of the passengers inside the stairs was 17% for those walking through the escalator and 83% for the passengers who remained standing. In addition, due to the high congestion of passengers, a distinction was made between the users who opted for the escalator and those who preferred the common escalator, with the observation that 15% opted for the common escalator and 85% opted for the escalator. On the other hand, it was possible to appreciate that the passengers at the moment of reaching the stairs did so in an orderly manner, generating two lines of flow to board the escalator (rows of 10 people long on average). It was possible to observe jamming; however, this was presented as a way of organizing the flow of passengers, which allowed them to board the escalator in a constant and orderly manner. An image of the situation can be seen in Figure 5, where it can be seen that the location of the stairs, next to the wall, influences the way in which passengers are organized to walk along the escalator. During observation, no passengers with reduced mobility were registered.

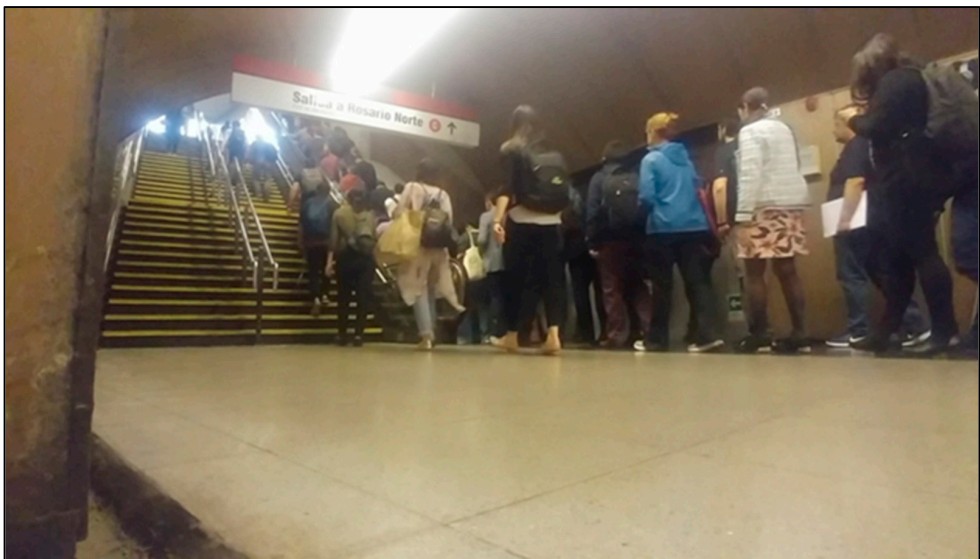

**Figure 5.** Passengers forming a queue when boarding the escalator in Manquehue station to exit the station (*Salida a Rosario Norte*).

- Tobalaba station: This station is of great importance since it is the combination of Line 4, which transports passengers from the southeast sector of the city, and Line 1, which is the central axis of the metro network, since it runs to central places characterized by a large number of workplaces, therefore, with high public attendance. The observations were made on a combination staircase that transfers passengers from the platform of Line 4 to Line 1, with it combining towards the San Pablo terminal station (see Figure 3a in Section 3.1). The observation was carried out for 15 min during the peak hour (8:00 a.m. to 8:15 a.m.). Similar to Manquehue station, the flow of passengers per minute on the escalator was measured, which turned out to be 50 passengers/min. However, within the variables that affected the behavior of the passengers, it was found that the length of the stair was 6 m, that is, the stair with the shortest length in this study. Regarding the location of the passengers on the stairs, in this observation, the users did not have the option of ascending the common stairs (since there is no common staircase), unlike Manquehue or Los Dominicos. However, it was observed that 16.5% walked along the escalator, while the rest (83.5%) kept standing. As for the boarding method of the passengers, disordered behavior was observed since the passengers came from two different directions, each one from a transfer corridor,

generating movements from the front and from the side. It is worth mentioning that the flow of passengers came from a side located 2 m before the entrance to the stairs, a sufficient distance to cause traffic jams when both flows converge at the entrance of the stairs. The traffic jam was not seen to be of great proportions but was limited to a maximum of four passengers trying to board the stairs at the same time. In this observation, as in the other escalators studied, no passengers with reduced mobility were found. It is important to note that in Tobalaba station, a tendency was observed in the location of passengers on the escalator when the capacity was reached, in which passengers kept a distance of one step from other passengers according to their location, left or right. This was also observed in Manquehue station due to the high number of passengers on the escalator. This behavior occurs due to the high agglomeration of passengers, reaching the capacity of the escalator.

### 4.2. Behavior Effect Using Detection Techniques

This section includes the analysis of the relationship between different conditions and passenger behaviors through passenger detection techniques and simulation tools. In the following, the results of the method are analyzed and discussed considering a pilot study at the Universidad de Chile station of the Santiago Metro (Line 1). The results are summarized in Table 2.

**Table 2.** Summary of observed behavior on escalators at Universidad de Chile station.

| Period of the Day | Passenger Flow [ped/min] | % Standing Passengers | % Walking Passengers | % Passengers Standing on the Right Side |
|---|---|---|---|---|
| A.M. (8:00–8:30 h) | 37 | 61.2% | 38.8% | 81% |
| P.M. (13:00–13:30 h) | 29 | 93.9% | 6.1% | 66% |

Four escalators were observed in Universidad de Chile station (see Figure 6). On each escalator, the stream of flows was analyzed using detection techniques. To this, the open-source software Datafromsky was used (https://datafromsky.com/, access date: 2 August 2023). An area of analysis was established on each escalator to count the number of passengers and their location on the escalator (e.g., walking or standing).

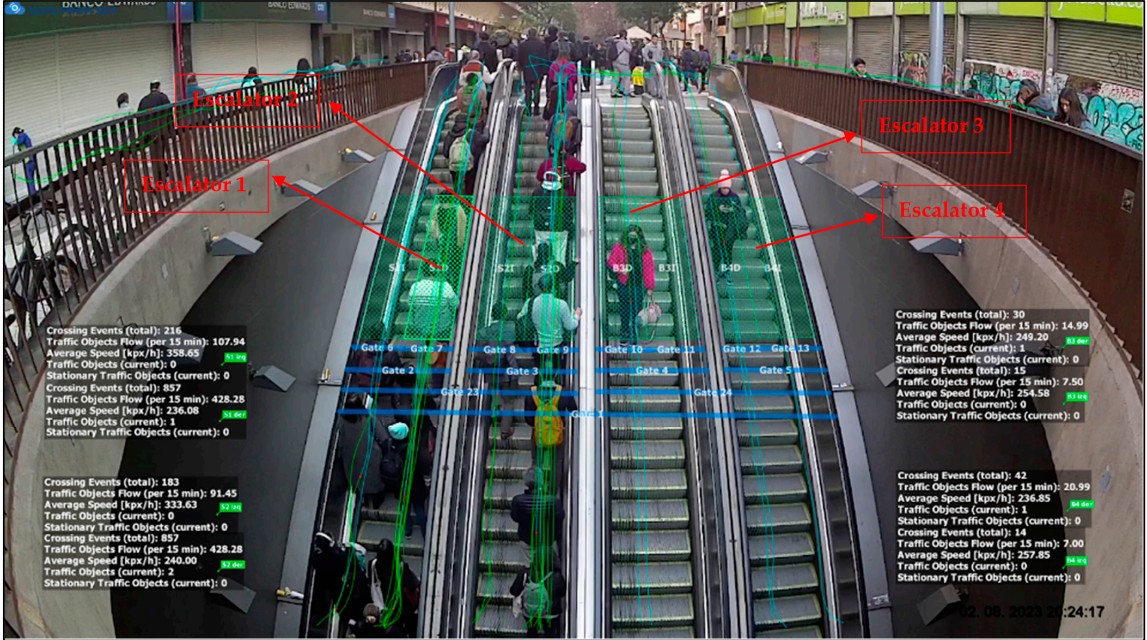

**Figure 6.** Screenshot of the escalators observed at Universidad de Chile station using detection techniques (in each escalator an area was defined to count passengers).

The results using the detection techniques during the peak and off-peak hours on the escalators at Universidad de Chile station are presented in Tables 3 and 4, respectively. The behavior of passengers on the escalator differs for each period of time. In the peak hour, passengers walk along the escalator, while during the off-peak hour, passengers stand on both sides of the steps, making it difficult for passengers to walk on the left. It can also be noticed that the previous behavior was obtained according to the difference between both periods of time regarding the speed and count of passengers, which is measured in kpx/h and passengers, respectively. The flow during the peak hour is mostly upstream (i.e., exiting the station), while during the off-peak hour, it is downstream (i.e., entering the station).

**Table 3.** Results using the detection techniques on the escalator at Universidad de Chile station during the peak hour.

| Flow Direction | Escalator | Location on the Steps | Average Speed [kpx/h] | Count [Passengers] |
|---|---|---|---|---|
| Downstream | 3 | Right | 221 | 30 |
| Downstream | 3 | Left | 224 | 15 |
| Downstream | 4 | Right | 220 | 42 |
| Downstream | 4 | Left | 227 | 14 |
| Upstream | 1 | Right | 233 | 857 |
| Upstream | 1 | Left | 413 | 216 |
| Upstream | 2 | Right | 238 | 857 |
| Upstream | 2 | Left | 405 | 183 |

**Table 4.** Results using the detection techniques on the escalator at Universidad de Chile station during the off-peak hour.

| Flow Direction | Escalator | Location on the Steps | Average Speed [kpx/h] | Count [Passengers] |
|---|---|---|---|---|
| Downstream | 3 | Right | 232 | 597 |
| Downstream | 3 | Left | 231 | 340 |
| Downstream | 4 | Right | 320 | 16 |
| Downstream | 4 | Left | 265 | 12 |
| Upstream | 1 | Right | 218 | 253 |
| Upstream | 1 | Left | 218 | 131 |
| Upstream | 2 | Right | 221 | 267 |
| Upstream | 2 | Left | 220 | 138 |

*4.3. Results Using the Simulation Framework*

Based on the pilot study at Universidad de Chile station, a simulation model was developed using Anylogic to simulate different escalator configuration scenarios to improve the efficiency of usage of the escalator. The simulation scenarios considered the following variables that affect the behavior of passengers:

- The location of the passengers on the escalator (e.g., right or left of each step).
- The way passengers reach the escalator (e.g., from the front or from the side).
- The number of lines of flow of passengers when boarding the escalator.
- The number of passengers (e.g., density).

The simulation model is available online via the web so that other researchers, practitioners, and students can use it [31]. In the simulation, three scenarios were tested:

- Scenario A: Passengers do not walk and they stand on both sides of the steps.
- Scenario B: Passengers walk from the left side and stand on the right side of the steps.
- Scenario C: Passengers walk on both sides of the steps.

The results are presented in Figure 7, in which the density of passengers is shown, with the blue representing the lowest density and the red representing the highest density on the escalator. It was noticed that Scenario C reached the lowest density level, as passengers

walked on both sides of the steps, reducing the agglomeration at the entrance of the escalator. On the other hand, Scenario B reached the highest density level, due to passengers walking only from the left and standing on the right of the steps. As a consequence, in Scenario B, a bottleneck was reached at the entrance of the escalator. In the case of Scenario A, a high density was observed on the escalator due to passengers standing on both sides of the steps.

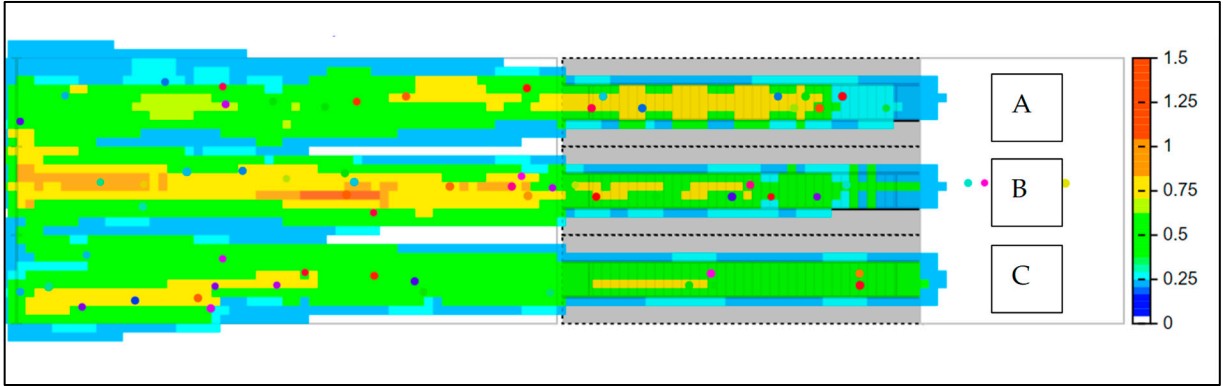

**Figure 7.** Density on the escalators using the simulation framework (each colored dot represents a passenger moving toward the escalator): (A) passengers do not walk and they stand on both sides of the steps; (B) passengers walk from the left side and stand on the right side of the steps; and (C) passengers walk on both sides of the steps.

In terms of travel time, the following figure should be considered. Figure 8 shows the time passengers spent accessing the escalator (e.g., before boarding) and the time spent on the escalator (e.g., walking or standing). It is interesting how the travel time changes for each scenario, in which Scenario C reached the lowest travel time, while Scenario A reached the highest.

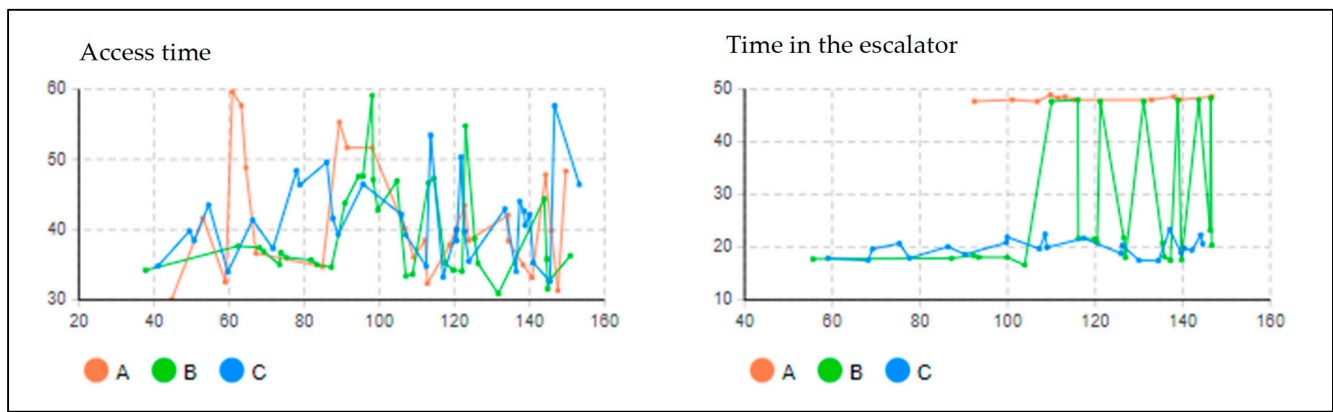

**Figure 8.** Travel time using the simulation framework: (A) passengers do not walk and they stand on both sides of the steps; (B) passengers walk from the left side and stand on the right side of the steps; and (C) passengers walk on both sides of the steps.

In addition, the interface is presented in Figure 9, in which passengers are represented as agents who are walking or standing on the escalator. The left escalator is Scenario A, the escalator in the middle represents Scenario B, and the right escalator is Scenario C.

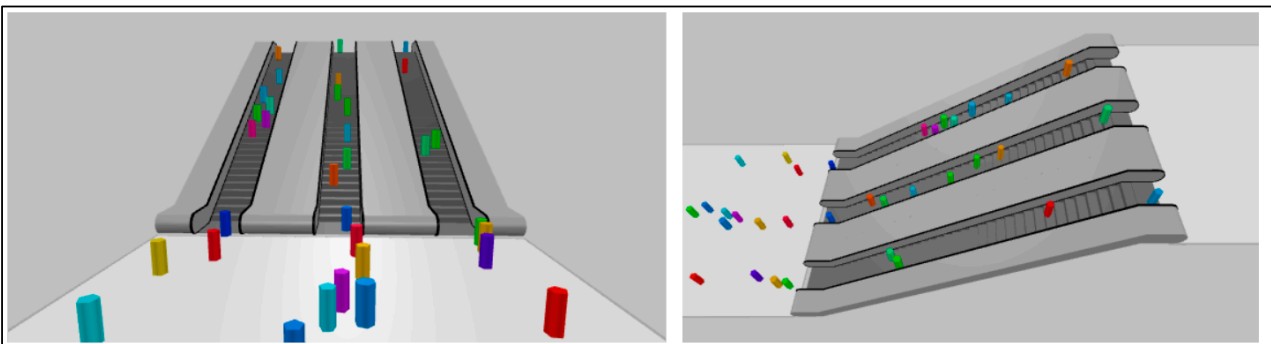

**Figure 9.** Interface using the simulation framework to understand the relationship between the different conditions and the behavior of passengers on the escalator (each colored dot represents a passenger moving toward the escalator).

The reason the simulation environment differs from the observations made at Universidad de Chile station is that the study is focused on evaluating the exit escalators of the station. The aim is to estimate the capacities based on their local dynamics and to determine what the required capacity should be in a scenario when passengers exit from the station. In the simulation scenarios, only exit escalators are needed to facilitate the flow of people. Although entrance escalators were observed in some scenarios, these are not the focus of this research and therefore were not included in the simulation environment.

### 4.4. Recommendations and Discussion

As part of the analysis, reviewing the variations that occur in terms of flow, it is evident that the higher the demand, the higher the density. However, when passengers walk along the escalator, a lower density is reached compared to the case when passengers are standing. The explanation for this lies in the behavior of passengers on the escalator and the use of the infrastructure. Similar to [10,14,18], it was found that the density, flow, and behavior of passengers are the most important variables that affect the capacity of escalators. When passengers stand on both sides of the escalator, they are located one step in between, maximizing the density. However, when passengers walk along the escalator, they use more space within it, which increases the effective area that is used by passengers, and therefore, the maximum density decreases because the same space is used by fewer passengers, which in turn reduces the discomfort of passengers.

In terms of travel time, it is clear that when passengers walk on the escalator, a lower evacuation time is obtained. This is caused by the passengers moving along the escalator reaching a higher flow. However, when the density along the escalator reaches a high level, it is more difficult to walk, and therefore, the time to evacuate the escalator could be reduced. The maximum flow obtained when passengers were standing on both sides of the steps was 60 passengers/min, reaching a density of 2 passengers/m$^2$ on the escalator. However, if passengers could walk along the escalator, the flow could reach up to 80 passengers/min, reaching a density of 1.5 passengers/m$^2$. This is summarized in the following tables which are classified according to Fruin's level of service (LOS) [13]. It is important to notice that the LOS = E is defined as capacity. As observed in Tables 5 and 6, the LOS is reported for different demand levels (the number of passengers boarding the escalator), in which the case of passengers walking along the escalator generated an improvement in the LOS compared to the situation when the passengers were standing on both sides of the steps.

**Table 5.** Level of service when passengers are standing on both sides of the escalator.

| Number of Passengers | Maximum Density [Passengers/m$^2$] | Maximum Flow [Passengers/min] | Average Speed of Passengers [m/s] | LOS Defined in [13] |
|---|---|---|---|---|
| 339 | 2.15 | 74.53 | 0.45 | F |
| 301 | 2.10 | 71.43 | 0.45 | F |
| 264 | 2.06 | 74.53 | 0.45 | F |
| 226 | 2.03 | 74.53 | 0.45 | F |
| 188 | 2.15 | 77.64 | 0.46 | F |
| 151 | 1.67 | 62.11 | 0.46 | E |
| 113 | 1.85 | 68.32 | 0.46 | E |
| 76 | 1.25 | 52.79 | 0.46 | D |
| 38 | 0.67 | 31.05 | 0.47 | C |

**Table 6.** Level of service when passengers are walking along the escalator.

| Number of Passengers | Maximum Density [Passengers/m$^2$] | Maximum Flow [Passengers/min] | Average Speed of Passengers [m/s] | LOS Defined in [13] |
|---|---|---|---|---|
| 339 | 1.55 | 80.74 | 0.60 | E |
| 301 | 1.48 | 68.32 | 0.59 | D |
| 264 | 1.48 | 80.74 | 0.61 | D |
| 226 | 1.58 | 74.53 | 0.60 | D |
| 188 | 1.47 | 74.53 | 0.66 | D |
| 151 | 1.34 | 65.22 | 0.60 | D |
| 113 | 1.10 | 55.90 | 0.56 | D |
| 76 | 0.57 | 31.05 | 0.65 | B |
| 38 | 0.51 | 27.95 | 0.65 | A |

As a consequence, to improve the usage and efficiency of this infrastructure, it is recommended to allow passengers to walk along the escalator and request passengers to stand to the right. But also, it is recommended to implement some crowd management measures to prevent passengers from forming a large queue at the entrance of the escalator (e.g., avoid a bottleneck by marking on the floor where to be located before boarding the escalator).

In summary, this study provides three main results which may be a valuable contribution to the literature:

- The use of video cameras and detection techniques. This study included the open-source software Datafromsky (https://datafromsky.com/, access date: 2 August 2023) which may be an alternative to Guo and Zhang [29], who analyzed a subway station using the Yolo artificial vision library. These researchers expanded the use of Datafromsky which has been widely used in traffic analysis [35].
- The behavior effect for the same escalator considering different periods of time. For example, the relationship between the trip's purpose and the type of passenger could be found. The results show that passengers may estimate the relevance of crowding to travel time, which could help to understand the relationship between congestion, comfort, and perceived security [36]. In addition, this study expands on what has been reported by other authors in terms of the perception of travel time and its impact on the well-being of passengers using the metro system [37].
- The development of a simulation framework. The use of simulation models in metro stations has been widely studied in the literature. The simulation framework of an urban railway station escalator using the software Anylogic expanded the results of previous studies reported in [22,31,32].

## 5. Conclusions

This research sought to analyze the escalator infrastructure in urban railway stations. To this, the behavior of passengers was studied through video cameras, detection techniques, and simulation tools, taking the Santiago Metro, Chile, as a pilot study.

Passenger behavior on escalators is influenced by cultural and environmental conditions. However, the passenger flow, density, and other cultural conditions can be quantified and evaluated on escalators, which makes it possible to determine their real capacity in a specific place.

Through a review of the state of the art and the observation of cases on escalators of the Santiago Metro, the variables that most influenced passenger behavior were identified. In this pilot study, it was observed that the behavior of passengers influenced the usage of the escalator infrastructure. When passengers board the escalator, they keep a separation step between each one and therefore respect the personal space around their bodies. A pattern of passenger distribution was also observed between those who decide to stand or those who walk along the escalator. In addition, it is relevant to understand the relationship between the trip's purpose and the type of passenger, which may affect the behavior of passengers on the escalator (e.g., peak vs. off-peak hours).

Based on these variables, a simulation framework was developed that uses an agent-based approach to reproduce the observed behavior in a simplified way. The use of a simulation framework has allowed us to understand the relationship between different conditions and the behavior of passengers, considering specific environmental and cultural conditions. In addition, it could be evaluated how the capacity differs from the one declared by the manufacturer and the value established in consideration of Fruin's level of service.

This paper is considered as a pilot study, and therefore, further studies are needed to analyze some situations that were observed partially in the field studies such as passengers starting to walk at the end of the escalator or depending on the empty space around them, among other situations which were not totally analyzed in this research. In addition, further research will consider applying this method in other stations of the Santiago Metro, which will help to increase the sample size to better compare the behavior on each escalator and include a statistical analysis.

Finally, this research aims to contribute to the state of the art and facilitate escalator capacity estimation by providing a user-friendly simulation framework. This framework will enable professionals and planners to more accurately assess the capacity of escalators and make informed decisions regarding their design and operation in different environments.

**Author Contributions:** Conceptualization, A.L. and S.S.; methodology, A.L. and S.S.; software, A.L.; validation, A.L. and S.S.; formal analysis, A.L., A.T. and S.S.; investigation, A.L. and S.S.; resources, S.S.; data curation, A.L., A.T. and S.S.; writing—original draft preparation, S.S.; writing—review and editing, A.L. and A.T.; visualization, A.L., A.T. and S.S.; supervision, A.L. and S.S.; project administration, S.S.; funding acquisition, S.S. All authors have read and agreed to the published version of the manuscript.

**Funding:** This research was funded by FONDECYT 11200012 and FONDEF id22i10018. Both research projects are from ANID, Chile.

**Institutional Review Board Statement:** The study was conducted according to the guidelines of the Declaration of Helsinki and approved by the Institutional Review Board (or Ethics Committee) of Universidad de Los Andes (protocol code: CEC202089, approved on 23 October 2020) and Pontificia Universidad Católica de Valparaíso (protocol code: BIOEPUCV-H 548-2022, approved on 3 October 2022).

**Informed Consent Statement:** Not applicable.

**Data Availability Statement:** Not applicable.

**Acknowledgments:** The author would like to thank the Santiago Metro for providing access for a site visit to the existing station during this research.

**Conflicts of Interest:** The authors declare no conflict of interest.

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
