# Peer review of "Analyzing Escalator Infrastructures: A Pilot Study in Santiago Metro"

_applsci, doi:10.3390/app132011117_

Round 1

Reviewer 1 Report

The authors observed the behaviors of escalator passenger through a method based on video cameras and simulated changes in performance depending on the behaviors. This research is useful when considering the installation of escalators. However, there are some unclear points in the presentation of observing result, and the reasons for selecting the experimental locations. The key concerns I have with the paper at this stage are as follows.

・The authors should show the overview of Santiago metro such as number of users and stations, lines, and length of rails. It is to give the reader an image of the scale of metro system.

・The observations at the three stations shown in Section 3 should be presented as a table. It would be easier for the reader to understand if the values presented and compared in the text (escalator length, structure, flow of people through the escalator, ratio of people walking to people standing) were presented as a table.  

・The reason for setting up the simulation environment shown in section 4 is unclear. You made detailed observations at each lane of the target station, which has four escalators. However, the simulation environment you created is different from all the corridors you observed because three of them go in the same direction. You need to explain the reason for your simulation setup.

Author Response

Thank you very much for reviewing this paper.

Please find attached the answers to the comments.

Reviewer 2 Report

The authors analyze the flow dynamics on three escalators in the Santiago metro system with intention to find behavioural changes that would improve throughput. 

1.     The analysis takes the whole of the data-gathering period and its video record to show overall trends in the data as well as probable causes. I have no problem with this as a first stage in the analysis, but it does not pin down the specific variables and their importance in the movement dynamics of the escalators. So we have a generalized observation on the data, which would be fine for the kind of recommendations envisaged by the authors. They are probably right in their inferences but there is uncertainty. The data are sufficient, it would appear, to submit to a more fine-grained analysis and using a different method of analysis as below. 

2.     There is a continuous record of movement, but it could be sampled by time to register a variety of situations undetectable in the generalized analysis. For example, upstream or downstream passengers will assess the situation ahead of them to decide whether it is worth walking or not on the escalator. When they see two people abreast, maybe they decide its not worth it, or decide to excuse themselves and pass. Similarly, a person standing on the left might disincentivize walking if the subsequent walker had to circumnavigate people in different positions. These situations might apply but only for those on the escalator or perhaps at the entrance area. These variables, along with total flow, might be crucial in the decisions and the resulting distribution of walking and standing.

3.     Sampling would allow inferential statistics so it would be possible to know the precise influence of each of several variables. This would strengthen the conclusions and allow generalization.

I have some more minor points:

1.     It is said in the Introduction that escalators are used to facilitate decongestion. Is this true? I could imagine a wide stairway that would accommodate more flow than one escalator. It could be that with an escalator present, there is more congestion at the base than would be the case on a stairway.

2.     At the beginning of 3. Method, the authors mention that the variables will be identified, but they should be more explicit right at the beginning. I realize the details are at the end of the section. In any event, I am recommending augmenting the list of variables.

3.     There are some essential references missing. For example,

Li Q, Ji C, Jia L, Qin Y. Effect of height on pedestrian route choice between stairs and escalator. Discrete Dynamics in Nature and Society. 2014:965305.

Cheung CY, Lam WHK. Pedestrian route choices between escalator and stairway in MTR stations. Journal of Transportation Engineering. 1998;124:277-285.

I saw only some incorrect prepositions.

Author Response

(The authors gave the same response as above.)

Reviewer 3 Report

1. Very limited literature review is cited. 

2. The methodology must be clear. In the present scenario it is very confusing.

3. The authors should incorporate a flow chart of the methodology of the overall study in the introduction section for further clearance of objective

4.  Proper discussion is needed on the results provided. 

5. How is this study better than the already present methods?

Author Response

(The authors gave the same response as above.)

Round 2

Reviewer 1 Report

The authors' revise clarified the overall structure of the paper and the assumptions of the experiment.

However, I would like to comment on the following points to make the paper better.

Ø The blue background and white text used in Figure 2 reduces readability. The authors should use black text on a white background.

Ø  Authors should describe the environment of the PC running Anylogic (CPU, OS, memory, Anylogic version) in Section 3.3.

Author Response

Thank you very much for the comments and for reviewing our paper. We included the following:

  • We changed the configuration of Figure 2 (in terms of colors). We use a white background and black text, which increases the readability.
  • We added the following sentence in section 3.3 to explain the environment of the PC running Anylogic: “It is important to mention that the model is running in Anylogic Cloud, version Anylogic Personal Learning Edition @{buildId} (Java version: 11.0.16), using the platform of Linux5.15.0-1028-aws 64-bit.”

Reviewer 2 Report

I appreciate the improvements, particularly in the literature review section. I maintain it should have been possible to conduct a statistical analysis, but at least the general findings are there.

Author Response

Thank you very much for your comments. As we mentioned in the reviewed version, this article is a pilot study, which will consider a detailed statistical analysis in further research. This was included in the conclusion section.

Reviewer 3 Report

thank you for the revisions.

Author Response

Thank you very much for reviewing our paper. We really appreciate all your comments, which were included in the previous version of the paper.